# Comparison of Measurement Possibilities by Non-Invasive Reflectometric Sensors and Invasive Probes

**Magdalena Paśnikowska-Łukaszuk** [1]**, Magda Wlazło-Ćwiklińska** [1]**,
Jarosław Zubrzycki** [2,*] **and Zbigniew Suchorab** [3]

1    Fundamentals of Technology Faculty, Lublin University of Technology, ul. Nadbystrzycka 38,
20-618 Lublin, Poland
2    Mechanical Engineering Faculty, Lublin University of Technology, ul. Nadbystrzycka 36,
20-618 Lublin, Poland
3    Faculty of Environmental Engineering, Lublin University of Technology, ul. Nadbystrzycka 40B,
20-618 Lublin, Poland
*    Correspondence: j.zubrzycki@pollub.pl

**Abstract:** The measurement of the moisture content of building materials is of key importance both in the process of building structures and in their subsequent operation. In engineering practice, indirect techniques of moisture measurement, mainly, resistance and capacitive, are the most popular. The main objective of this research work was to compare the classic TDR measurement technique to the non-invasive, surface TDR sensors. Moisture measurements were carried out on samples made of cellular concrete with density class of 400 and 600. These samples were moist to various degrees, from 0 to 69% (400 c.c.) and from 0 to 55 (600 c.c.). For each sample, five measurements were carried out. Both the RMSE and the expanded uncertainty values were more favorable for the TDR FP/mts probe and were consistent with the literature data. Compared to them, the measurement result for the 400 c.c. samples with the S1 probe was 154.6%, and that with the S2 probe was 87.03% of the values obtained with the invasive probe. When measuring the 600 c.c. samples, we found values of 122.16% for S1 and of 120.1% for S2 of those obtained with the invasive probe. The use of surface TDR sensors provided an easy and quick measurement without damaging the surface and structure of the tested material, as there was no need to introduce the probe actuators inside the tested material.

**Keywords:** TDR sensor; moisture; porous materials; calibration model





## 1. Introduction

Measurement research has been refined over the years by developing various data acquisition methods. The study of environmental factors involves many measurement aspects. One of such factors is moisture, which affects the building materials, but also the quality of life of the building users [1]. The moisture of building materials depends primarily on their absorption properties, as well as on the operating conditions and on independent external factors, which are often difficult to detect also due to the improper construction of objects, faults or the lack of anti-moisture insulation [2,3]. Moisture tests are also carried out in order to determine all the factors that lead to the multiplication of fungi, especially molds, that threaten human health [2]. Moisture can cause physical, chemical and biological damage [4]. A moistened material is subject to faster corrosion. With increased fungal infection, it can also be a substrate for the development of other microorganisms, additionally losing its thermal insulation properties [5]. If the moisture condensed in the porous space freezes, the structure of the material is destroyed. The processes of freezing and thawing lower the strength parameters of building materials [6]. An important element when examining building structures is the assessment of moisture parameters and the identification of harmful salts that may be present in the building material in various proportions [7]. The sources of moisture are often unnoticeable and may

be hidden under the insulation or located deep in the building material, which may make their detection difficult [8,9]. Capillary water located deep in a wall is especially harmful. This water destroys the wall from the inside and is invisible on the wall surface, unlike condensation water that appears on the surface and is the result of improper ventilation of a room [10].

For the quick detection of moisture in a wall, indirect techniques are the most useful, enabling to estimate the moisture of a partition on the basis of other physical parameters, the values of which indirectly depend on moisture. Most often, these are electrical techniques based on the measurement of electrical conductivity or dielectric permittivity [11]. An advantageous feature of the electrical methods is that most often these methods are non-invasive and allow the testing of building materials without disturbing their structure and shape [12–14]. The basic electrical method is the resistance technique, which consists in measuring the conductivity or electrical resistance of the tested materials in porous materials depending on the degree of their moisture [15]. The measures based on this method are most often calibrated to the mass moisture of the material [16] and are most often used to determine the moisture of construction wood. However, the environmental parameters of walls or partitions are commonly determined using other methods [17,18]. These include the capacitive Frequency Domain (FD) method, which is considered to be better than the resistance technique. The FD method consists in measuring the capacitance of a properly constructed capacitor with alternating voltage. In FD methods, the conductive electrodes in the area of the test medium are treated as capacitor plates. The dielectric of this capacitor is the material to be measured. The value of the apparent permittivity of the material affects the capacity of the capacitor thus formed. The measurement of this capacity allows the assessment of the material's moisture [19,20]. Another non-invasive method, i.e., microwaves, allows measuring the phase shift and the degree of electromagnetic wave attenuation in the tested material, which allows the further determination of the moisture content. With the help of microwave moisture measurements, it is possible to determine the degree of moisture in building partitions inside their structure and on their surface. Microwave radiation is absorbed by matter through ionic conductivity or through the phenomenon of dielectric losses resulting from dipole polarization [21–24]. Water is a dipole which, while appearing in the structure of another material, still maintains the asymmetric nature of its molecules [25].

A separate group of measurement methods are the direct methods that enable to measure the presence of water. These are mostly invasive methods that interfere with the structure or shape of the tested element. The invasive measurement method which is most frequently used to determine the moisture in walls is the gravimetric method [26]. The basic instrument used in this method is a moisture analyzer, which can be used to test the material taken from a wall. The test material is weighed and then dried and reweighed in order to determine the weight differences [27]. It is a method that requires the destruction of the material structure.

Destructive methods also include the carbide method (CM) [28]. This is an indirect method that also involves sampling and testing with a CM hygrometer using the chemical process of calcium carbide decomposition by water. During decomposition, acetylene gas is released, which causes an increase in pressure in the device [29]. Other invasive methods supporting the moisture measurement process also include measurements with a Peltier probe, which determines a sample's suction potential in a given moisture state. Instead of the material moisture, the psychrometric Peltier probe measures the water potential, which is an indicator of the ability of a porous material to bind water [30]. Thus, the relative moisture of a porous material can be calculated [31]. In order to conduct invasive psychrometric measurements, the probe must be placed in the tested material by drilling a hole, which, if the material is loose, can be done by pressing the probe [31]. It is a standard form of measurement, additionally used to determine parameters that can support the process of measuring the moisture content of building materials.

A measuring technique that can combine the advantages of destructive and non-invasive techniques is Time Domain Reflectometry (TDR), which uses the measurement of the propagation time of a short electromagnetic pulse in a material sample to determine the apparent permittivity of porous materials, which is strongly dependent on the water presence, i.e., material moisture [21,32–41]. The apparent permittivity of the medium determines the velocity of signal propagation along the waveguide. The TDR sensor can also be used for the non-invasive monitoring of changes in moisture content in rigid porous materials [42]. The design of a TDR sensor determines the shape of the reflected signal, and the accuracy of the propagation time measurement depends on its design details [43]. Common applications of the TDR technique are based on the use of typical invasive probes introduced into the tested porous medium; this technique is most often used in soil science. With reference to building materials, its use involves the destruction of samples or walls [44]. Therefore, in the case of hard and rigid media, which include building materials, surface sensors are an alternative to traditional probes [42,45,46]. In the measurement of the basic environmental factors of building partitions, which include moisture, the TDR technique should be treated as a method at the stage of implementation for the purpose of testing walls. In order to improve this measurement technology, the obtained test results should be properly analyzed using typical invasive and non-invasive sensors and compared to the reference values obtained by direct testing [11]. The aim of this article was to determine the possibility of measuring moisture in building materials with non-invasive TDR sensors and to compare the measuring potential of this method to that of the TDR method using traditional invasive sensors.

## 2. Materials and Methods

A set of cellular concrete samples was used for the measurements. The devices that were utilized for sample preparation were the following: a 06-DZ-3BC laboratory oven (Chemland, Stargard, Poland), an SBS-LW-3000N laboratory scale (Steinberg Systems, Zielona Góra, Poland), TDR equipment including a LOM laboratory multimeter (ETest, Lublin, Poland), traditional TDR FP/mts probes described in detail in Section 2.1.1 (ETest, Lublin, Poland), TDR surface sensors (manufactured in Lublin University of Technology) described in detail in Section 2.1.2 and a Personal Computer serving for TDR multimeter control and data management.

### 2.1. Materials

#### 2.1.1. TDR Meter Description

The measurements were conducted with a TDR multimeter emitting a needle peak signal with rise time equal to 300 ps, produced by ETest manufacturer, Lublin, Poland. The emitted signal propagated along the coaxial cable to the sensor where reflections occurred on the characteristic points of the propagation line. Those reflections derived from both the beginning of the sensor and its termination and served as measurement markers. The time differences between those reflections were read by the TDR meter and could be automatically or manually recalculated into the apparent permittivity value that depends on a material's moisture.

#### 2.1.2. FP/Mts Sensor

The FP/mts sensor presented in Figure 1 was intentionally designed for the in situ evaluation of soil moisture but for several years has been successfully applied for the moisture evaluation of soft building materials [42].

Its main functional elements are the two 10 cm long sharpened acid-resistant steel rods (2 mm in diameter, separated by 14 mm), a sensor support made from a section of a PVC tube of 2 cm outer diameter and length, and a coaxial cable of length from 1.5 to 6 m, from the sensor to the terminating SMA connector.

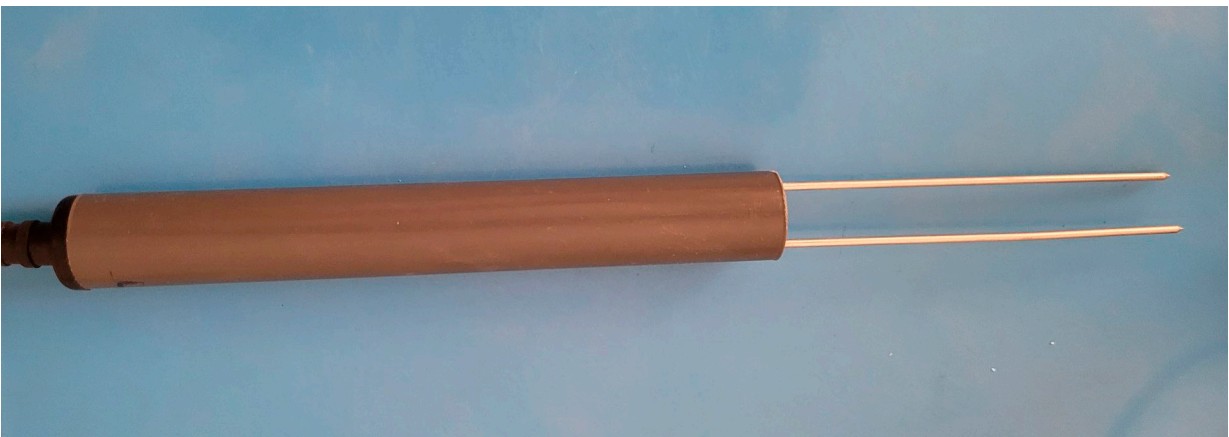

**Figure 1.** FP/mts TDR sensor applied in this research (ETest, Lublin, Poland).

### 2.1.3. Non-Invasive TDR Sensors

For the experiment, two non-invasive TDR sensors manufactured in Lublin University of Technology were applied. The sensors were previously described in articles by Suchorab et al. [42,43].

### 2.1.4. S1 Non-Invasive Sensor

The S1 non-invasive sensor (Figure 2a,b) is made of black polyoxymethylene, characterized by an apparent permittivity value of 3.8 [-] [47]. The length of the measuring probe is 200 mm, and its width is 50 mm. The measuring rods are made of a brass flat bar with a cross section of $2 \times 10$ mm. The sensor communicated with the TDR meter via an angled BNC connector that was soldered to the printed circuit board that linked the measuring rods to the connector. In the design of this sensor, a flat bar was placed in a dielectric.

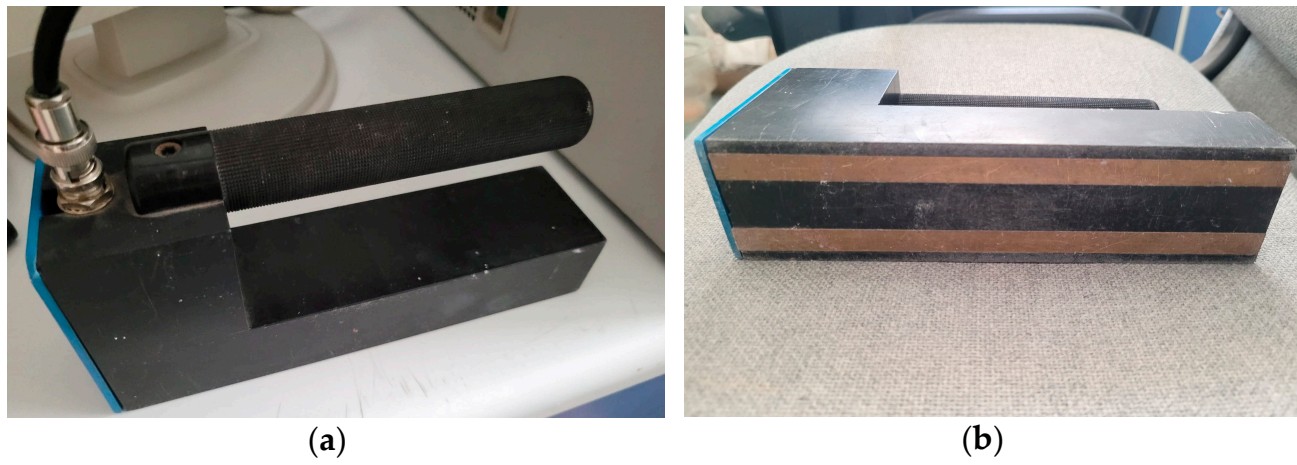

(**a**)　　　　　　　　　　　　　　　　(**b**)

**Figure 2.** S1 non-invasive TDR sensor [42]. (**a**) Isometric view, (**b**) view from the bottom.

### 2.1.5. S2 Non-Invasive Sensor

The S2 sensor shown in Figure 3a,b is similar in construction and made of the same material as the S1 sensor. Its length is 200 mm, and its width is 100 mm. As in the case of the S1 sensor, the waveguides of the probes were made of a brass flat bar with a cross section of 2 mm $\times$ 10 mm.

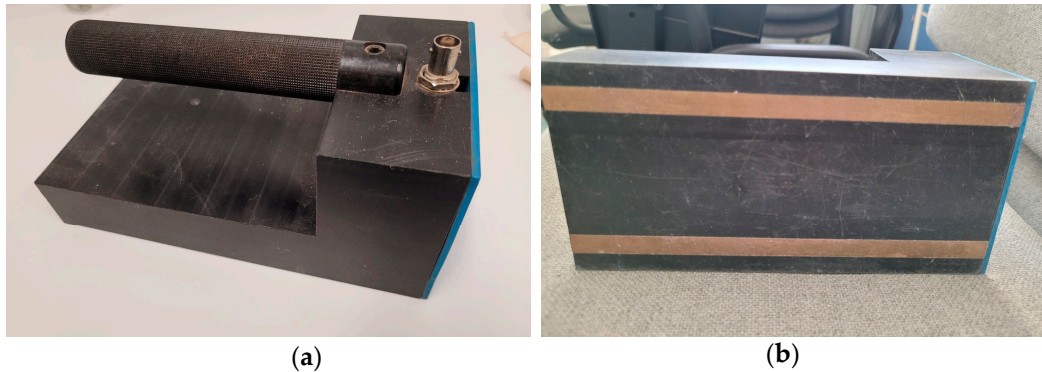

(**a**)　　　　　　　　　　　　　　　　　　　(**b**)

**Figure 3.** S2 non-invasive TDR sensor [42]. (**a**) Isometric view, (**b**) view from the bottom.

### 2.2. Description of the Tested Material

Cellular concrete was used for the tests as the building material. In the presented research, samples of 400 kg/m$^3$ and 600 kg/m$^3$ of cellular concrete were used.

### 2.3. Samples and Preparations

2.3.1. Samples for Invasive Measurements

Due to the FP/mts invasive probe geometry and its sensitivity range (ETest, Lublin, Poland), a set of samples of cellular concrete was prepared. The samples with dimensions equal to 5 × 5 × 12 cm were prepared in the amount of 40 pieces in the case of the 400 kg/m$^3$ cellular concrete (400 c.c.) and of 35 pieces in the case of the 600 kg/m$^3$ cellular concrete (600 c.c.). They were dried to a constant mass at a temperature of 105 °C. Then, the samples were saturated to the desired moisture by dosing the appropriate amount of water until the expected moisture was obtained, in steps of 10% by mass, until the state of full saturation (69% mass in the case of cellular concrete 400, and 55% mass in the case of cellular concrete 600).

2.3.2. Samples for Non-Invasive Measurements

The dimensions of the samples used in non-invasive research were 220 mm × 120 mm × 40 mm. Similarly to the samples for invasive measurements, they were dried to a constant mass and gradually moistened to achieve 69% and 55% of moisture. The samples were weighed to check the moisture status. Then, the samples were examined with the non-invasive S1 sensor and subsequently with the S2 sensor in order to obtain the measurement results.

### 2.4. Methods

Description of the Measurement Procedure

The research consisted in measuring the apparent permittivity of the material with different moisture values. The measurements were made on dry samples (5 readouts for statistical purposes) and then on samples of increasing moisture until saturation. The tests were carried out under constant conditions of temperature (20 °C) and relative air moisture (50%).

In the case of the invasive FP/mts probes, they were introduced into the structure of the tested material.

In the case of the non-invasive sensors, the measurements were carried out using the touch method on the samples with dimensions of 220 mm × 120 mm × 40 mm.

### 2.5. Data Analysis Method

As a result of this research, we obtained measurements based on the relationship between the apparent permittivity readings obtained with the TDR sensors and the mass moisture content of the material of the tested samples. The uncertainty of the measurements

was assessed with the use of appropriate regression models. A polynomial relationship was used for the calibration based on the formula:

$$\hat{w} = \beta_1 \cdot \varepsilon^2 + \beta_2 \cdot \varepsilon + \beta_3 + \in \tag{1}$$

where $\hat{w}$ is the mass moisture value estimated using a polynomial model [%mass], $\varepsilon$ is the apparent permittivity measured using TDR, $\epsilon$ is the random error of normal distribution, $\beta_{1-3}$ are the model estimators.

The parameters defining the measurement uncertainty of the methods are the mean-square error (RMSE) and the coefficient of determination $R^2$, which allows for the assessment of the regression model adaptation and the quality of the model fit in relation to the measured data.

Additionally, an extended assessment of the measurement uncertainty (standard and extended uncertainty) of the invasive FP/mts TDR sensors and the non-invasive S1 and S2 sensors was performed based on the GUM guidelines [48].

The uncertainty assessment included the complex uncertainty assessment, which is a combination of two types of uncertainty, i.e., type A, that is statistical uncertainty depending on the quality of the adopted model fit, and type B, which depends on the uncertainty and resolution of the individual device. With a complex standard uncertainty, the expanded measurement uncertainty was estimated [43].

For the applied measurement methods, type B uncertainties were much lower than type A uncertainties and were ignored in the calculations; therefore, the following elements were taken into account for the estimation of uncertainty: the estimators $\beta_0$, $\beta_1$, $\beta_2$ and the relative permeability ($\varepsilon$):

$$w = f(\beta_1, \beta_2, \beta_3, \varepsilon) \tag{2}$$

The composite standard uncertainty of the measurement mentioned above, which included both type A and type B uncertainties, can be described by the formula:

$$u_C(w) = \sqrt{\left(\frac{\partial w}{\partial \varepsilon} u(\varepsilon)\right)^2 + \sum_{i=0}^{2}\left(\frac{\partial w}{\partial \beta_i} u(\beta_i)\right)^2 + 2\sum_{i=0}^{2}\sum_{j=i+1}^{2}\frac{\partial w}{\partial \beta_i}\frac{\partial w}{\partial \beta_j}u(\beta_i, \beta_j)} \tag{3}$$

The expanded uncertainty can be described with the following formula [47].

$$U(w) = k_p \cdot u_c(w) \tag{4}$$

where $k_p$ is the coverage factor that depends on the number of degrees of freedom, whose value is approximately 2.

### 3. Results

Tables 1–6 show the permittivity readings by the different sensor types (i.e., invasive FP/mts and non-invasive S1 and S2) for the two classes of cellular concrete with different bulk moisture (400 and 600). The first column of each table shows the mass moisture $w$, and the following columns show the individual moisture readings for each tested sample, $\varepsilon_1$–$\varepsilon_5$.

**Table 1.** Results of the apparent permittivity $\varepsilon$ measurements with the FP/mts invasive probe for different values of material moisture $w$; 400 c.c. sample.

| Moisture $w$ [%] | Apparent Permittivity $\varepsilon$ [-] | | | | |
|---|---|---|---|---|---|
| | $\varepsilon_1$ | $\varepsilon_2$ | $\varepsilon_3$ | $\varepsilon_4$ | $\varepsilon_5$ |
| 0 | 1.18 | 1.16 | 1.14 | 1.2 | 1.18 |
| 10 | 2.14 | 2.01 | 2.22 | 1.84 | 2.41 |
| 20 | 5.31 | 5.82 | 5.09 | 5.42 | 5.47 |
| 30 | 7.17 | 6.87 | 7.55 | 7.25 | 7.02 |

**Table 1.** *Cont.*

| Moisture $w$ [%] | Apparent Permittivity $\varepsilon$ [-] | | | | |
|---|---|---|---|---|---|
| | $\varepsilon_1$ | $\varepsilon_2$ | $\varepsilon_3$ | $\varepsilon_4$ | $\varepsilon_5$ |
| 40 | 9.32 | 9.47 | 9.37 | 9.27 | 9.07 |
| 50 | 12.54 | 11.91 | 12.35 | 12.01 | 12.78 |
| 60 | 15.55 | 15.38 | 14.84 | 15.83 | 15.26 |
| 69 | 20.35 | 20.27 | 20.33 | 20.55 | 20.11 |

**Table 2.** Results of the apparent permittivity $\varepsilon$ measurements with the S1 surface sensor for different values of moisture $w$; 400 c.c. sample.

| Moisture $w$ [%] | Apparent Permittivity $\varepsilon$ [-] | | | | |
|---|---|---|---|---|---|
| | $\varepsilon_1$ | $\varepsilon_2$ | $\varepsilon_3$ | $\varepsilon_4$ | $\varepsilon_5$ |
| 0 | 3.63 | 3.73 | 3.54 | 3.44 | 3.65 |
| 10 | 4.06 | 3.97 | 4.05 | 3.921 | 4.17 |
| 20 | 5.00 | 5.09 | 5.05 | 5.09 | 5.09 |
| 30 | 6.05 | 6.00 | 6.05 | 5.67 | 5.43 |
| 40 | 7.54 | 7.53 | 7.42 | 7.77 | 7.48 |
| 50 | 7.98 | 7.54 | 7.81 | 7.77 | 7.95 |
| 60 | 8.94 | 9.13 | 9.07 | 8.88 | 9.21 |
| 69 | 12.92 | 13.16 | 12.92 | 12.64 | 12.71 |

**Table 3.** Results of the apparent permittivity $\varepsilon$ measurements with the S2 surface sensor for different values of moisture $w$; 400 c.c. sample.

| Moisture $w$ [%] | Apparent Permittivity $\varepsilon$ [-] | | | | |
|---|---|---|---|---|---|
| | $\varepsilon_1$ | $\varepsilon_2$ | $\varepsilon_3$ | $\varepsilon_4$ | $\varepsilon_5$ |
| 0 | 3.56 | 3.59 | 3.67 | 3.79 | 3.71 |
| 10 | 4.18 | 3.88 | 4.01 | 4.25 | 4.28 |
| 20 | 5.38 | 5.16 | 5.34 | 5.02 | 5.20 |
| 30 | 6.04 | 6.24 | 5.90 | 6.38 | 6.17 |
| 40 | 7.26 | 7.30 | 6.84 | 6.69 | 7.10 |
| 50 | 8.31 | 7.86 | 7.69 | 7.69 | 7.96 |
| 60 | 9.62 | 9.56 | 10.22 | 10.22 | 10.09 |
| 69 | 12.25 | 11.86 | 11.53 | 11.67 | 11.88 |

**Table 4.** Results of the apparent permittivity $\varepsilon$ measurements with the FP/mts invasive probe for different values of moisture w; 600 c.c. sample.

| Moisture $w$ [%] | Apparent Permittivity $\varepsilon$ [-] | | | | |
|---|---|---|---|---|---|
| | $\varepsilon_1$ | $\varepsilon_2$ | $\varepsilon_3$ | $\varepsilon_4$ | $\varepsilon_5$ |
| 0 | 1.06 | 1.07 | 1.11 | 1.22 | 1.04 |
| 10 | 2.75 | 2.38 | 2.91 | 2.83 | 2.64 |
| 20 | 5.48 | 5.21 | 5.84 | 5.49 | 5.65 |
| 30 | 6.74 | 7.20 | 6.91 | 6.59 | 7.07 |
| 40 | 11.45 | 11.31 | 10.94 | 11.24 | 11.67 |
| 50 | 16.57 | 16.24 | 16.97 | 16.85 | 16.78 |
| 55 | 19.20 | 20.41 | 19.48 | 20.30 | 20.21 |

**Table 5.** Results of the apparent permittivity $\varepsilon$ measurements with the S1 surface sensor for different values of moisture $w$; 600 c.c. sample.

| Moisture $w$ [%] | Apparent Permittivity $\varepsilon$ [-] | | | | |
|:---:|:---:|:---:|:---:|:---:|:---:|
| | $\varepsilon_1$ | $\varepsilon_2$ | $\varepsilon_3$ | $\varepsilon_4$ | $\varepsilon_5$ |
| 0 | 3.33 | 3.42 | 3.25 | 3.32 | 3.50 |
| 10 | 3.82 | 4.05 | 3.94 | 3.88 | 3.77 |
| 20 | 5.27 | 5.44 | 5.28 | 5.29 | 5.20 |
| 30 | 6.27 | 6.58 | 6.50 | 6.39 | 6.58 |
| 40 | 9.05 | 8.79 | 8.58 | 8.83 | 8.85 |
| 50 | 9.78 | 10.27 | 9.82 | 10.05 | 9.84 |
| 55 | 12.05 | 12.25 | 12.31 | 12.62 | 12.35 |

**Table 6.** Results of the apparent permittivity $\varepsilon$ measurements with the S2 surface sensor for different values of moisture $w$; 600 c.c. sample.

| Moisture $w$ [%] | Apparent Permittivity $\varepsilon$ [-] | | | | |
|:---:|:---:|:---:|:---:|:---:|:---:|
| | $\varepsilon_1$ | $\varepsilon_2$ | $\varepsilon_3$ | $\varepsilon_4$ | $\varepsilon_5$ |
| 0 | 3.11 | 3.32 | 3.27 | 3.31 | 3.43 |
| 10 | 3.72 | 3.76 | 3.87 | 3.94 | 3.84 |
| 20 | 5.11 | 5.24 | 5.42 | 5.30 | 5.36 |
| 30 | 6.75 | 6.81 | 6.99 | 6.85 | 6.86 |
| 40 | 8.82 | 8.96 | 9.10 | 9.12 | 8.96 |
| 50 | 10.53 | 10.28 | 10.05 | 9.96 | 10.18 |
| 55 | 12.65 | 12.49 | 12.88 | 12.72 | 12.57 |

## 4. Discussion

### 4.1. Regression Model

The graphs in Figure 4 present the relationships obtained on the basis of the results presented in Tables 1–6. The graphs show the average permittivity readings from the five measurements for all examined moisture contents and the confidence intervals.

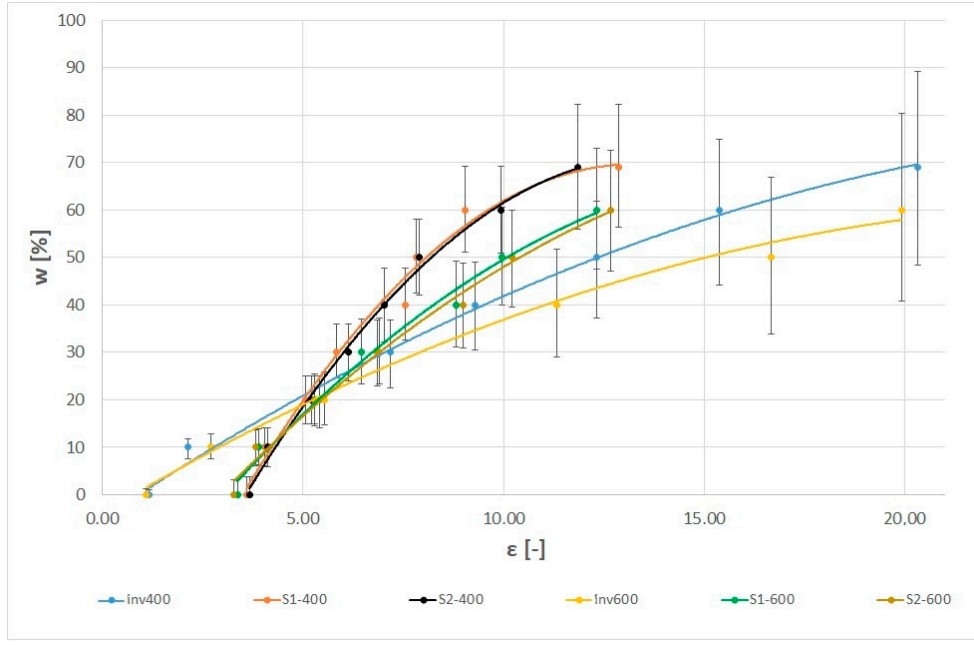

**Figure 4.** Relationship between the apparent permittivity reads determined with the use of the FP/mts invasive probe and the non-invasive probe and the moisture content of class 400 c.c. and 600 c.c. samples.

The graphs in Figure 4 confirmed the relationship between the apparent permittivity and the moisture content of the material. The effect of the apparent bulk density of the material was not high, and the different regression courses resulted from the fact that the two materials had different water absorption capacity, i.e., a maximum of 55% for the aerated concrete 600 and a maximum of 69% for the aerated concrete 400, which was a consequence of the different structure of the porous medium.

There were differences in the permittivity readings between the invasive probes and the surface sensors, which resulted from their construction and the type of contact between the measuring element and the material. In the case of the invasive probes, the measuring rods were inserted into the material, whereas in the case of the surface sensors, they were in contact with the surface of the material. The invasive probes read the permittivity of the tested material, and the apparent permittivity read by the sensors S1 and S2 was the average value of the apparent permittivity of the tested material as well as of the housing from which the sensor was made. As a consequence, in the case of the invasive sensors for dry material, the average apparent permittivity value was 1.17 [-] for the cellular concrete 400 and 1.1 [-] for the cellular concrete 600. However, in the case of the S1 and S2 probes, they were 3.60 [-] and 3.66 [-] for the concrete 400 and 3.36 [-] and 3.29 [-] for the concrete 600. For the maximum saturation of the 400 cellular concrete sample, the apparent permittivity result when measured with an invasive probe exceeded 20 [-].

On the basis of the measurements, the obtained data and the adopted polynomial regression model described by formula (1), the values of the $\beta_1$ $\beta_2$ and $\beta_3$ estimators were calculated for the individual sensor models and the classes of cellular concrete. These values are summarized in Tables 7 and 8.

**Table 7.** Estimator values of the adopted calibration models for the measuring probes used (400 c.c. sample).

| Sensor | $\beta_1$ | $\beta_2$ | $\beta_3$ | $R^2$ | F Statistic | RSE [%mass] | RMSE [%mass] |
|---|---|---|---|---|---|---|---|
| Invasive | −0.098 ** | 5.660 *** | −5.005 | 0.994 | 419.593 *** (df = 2; 5) | 2.212 | 1.748806 |
| Non-invasive (S1) | −0.733 ** | 19.419 *** | −58.949 *** | 0.986 | 174.069 *** (df = 2; 5) | 3.42 | 2.703889 |
| Non-invasive (S2) | −0.684 ** | 18.86 *** | −58.64 *** | 0.996 | 555.122 *** (df = 2; 5) | 1.925 | 1.521514 |

df—degree of freedom, *p*—critical level of significance (* $p < 0.05$; ** $p < 0.01$; *** $p < 0.001$).

**Table 8.** Estimator values of the adopted calibration models of the measuring probes used (600 c.c. sample).

| Sensor | $\beta_1$ | $\beta_2$ | $\beta_3$ | $R^2$ | F Statistic | RSE [%mass] | RMSE [%mass] |
|---|---|---|---|---|---|---|---|
| Invasive | −0.127 ** | 5.482 *** | −4.827 | 0.993 | 286.774 *** (df = 2; 4) | 2.089 | 1.579467 |
| Non-invasive (S1) | −0.458 * | 13.056 ** | −36.376 ** | 0.990 | 191.525 *** (df = 2; 4) | 2.552 | 1.929395 |
| Non-invasive (S2) | −0.379 * | 11.670 ** | −31.802 ** | 0.990 | 555.122 *** (df = 2; 5) | 2.509 | 1.896724 |

df—degree of freedom, *p*—critical level of significance (* $p < 0.05$; ** $p < 0.01$; *** $p < 0.001$).

In Table 7, it can be seen that the measurement error expressed as RMSE in the case of 400 c.c. was the highest when using the S1 non-invasive sensor. On the other hand, the values of the determination coefficient $R^2$ were similar for all analyzed sensors and exceeded the value of 0.98, which proved a very good fit of the adopted model to the dependence tested. In turn, in the case of class 600 c.c., the RMSE error values presented in

Table 8 were the highest for the invasive measurement by the FP/mts probe. As in the case of the 400 c.c. sample, the values of the $R^2$ coefficient were similar in all measurements for the class 600 c.c. samples and, in each case, exceeded 0.99.

In the case of the cellular concrete 400, the critical significance levels of the $\beta_2$ and $\beta_3$ estimators assumed a value lower than 0.001 for all sensors, which means that they were statistically significant. In turn, the critical significance levels of $\beta_1$ for all sensors assumed a value of less than 0.01. In the case of the aerated concrete 600, the $\beta_2$ and $\beta_3$ estimators were less than 0.1, and the $\beta_1$ estimator was less than 0.05. This means that their significance levels were lower for all sensors. On the other hand, the analysis of the F statistic ($p < 0.001$) in all cases confirmed the statistical significance of the adopted regression models.

### 4.2. Uncertainty Analysis

The results obtained from the measurements made it possible to describe the relationship between the apparent permittivity and the moisture content of the material with the second-grade polynomial regression model according to formula (2) and to determine the quality of the model fit to the obtained data using the determination coefficients $R^2$, as well as the standard errors of measurement expressed as Residual Standard Error (RSE) and Root-Mean-Square Error (RMSE). Additionally, based on the GUM [49], the analysis of the measurement uncertainty and expanded uncertainty in the material moisture function was performed. The calculations were made on the basis of formulas (3) and (4). The obtained results are presented in the form of graphs in Figure 5.

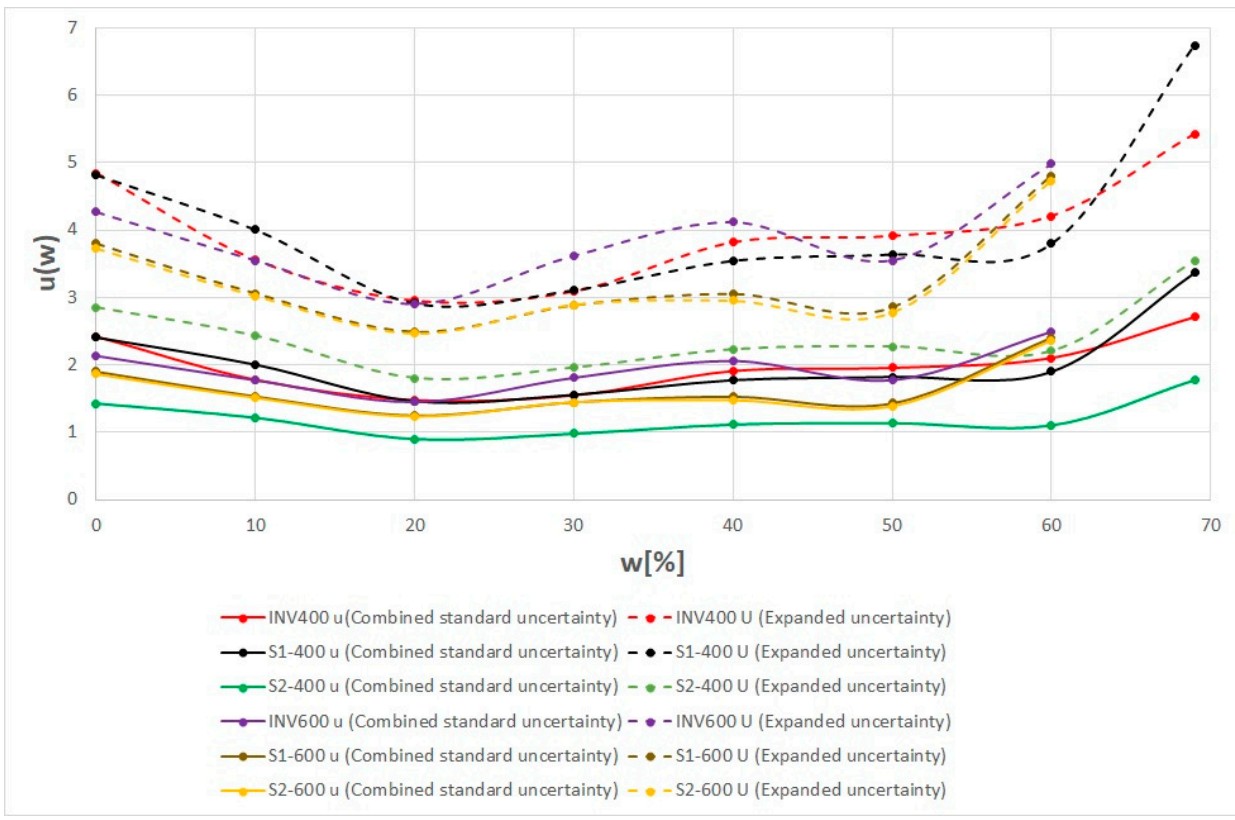

**Figure 5.** Uncertainty dependence on material moisture: invasive, S1 and S2 sensor, 400 and 600 c.c. samples.

### 4.3. Discussion of the Calibration Results and Uncertainty Calculations

In the case of dry samples and a moisture content below 0.05 cm$^3$/cm$^3$, the apparent permittivity determined by means of the surface sensors ranged from 3 to 4. This was a consequence of the value of the apparent permittivity of the solid phase of the tested

material and the apparent permittivity of polyoxymethylene of 3.8 [-] [43]. In the higher moisture ranges, the TDR surface sensor readings showed higher moisture values than the conventional invasive probe.

This was also confirmed by the statistical characteristics of the model used, mainly, the coefficient of determination ($R^2$) equal to 0.986 and 0.996 for the S1 and S2 TDR surface sensors and to 0.988 for the invasive probe. The RSE values were equal to 1.925 and 3.42 for the non-invasive TDR sensors and to 3.183 for the invasive probe. Similar observations were made in the case of the RMSE. It was equal to 2.70- and 1.52 for the TDR surface sensors and to 2.52 for the invasive probe. The RSE and RSME values were lower for the non-invasive S2 probe, while $R^2$ was the highest for this probe among the tested sensors [42,49].

Due to the fact that in the measurements of building materials and partitions the most frequently used quantity is the mass moisture (in %), calibration models were developed for these units. Unfortunately, in the case of the TDR technique, most of the available literature expresses a material's moisture content as volumetric moisture; therefore, for the purposes of literature discussion, the obtained values were converted to $cm^3/cm^3$ according to the formula presented in [11,42].

Tables 7 and 8 show that the RMSE values obtained for all probe models ranged from 1.521% mass to 2.7% mass (0.0091 to 0.0162 $cm^3/cm^3$) for the cellular concrete 400 and from 1.579% mass to 1.929% mass (0.0063 to 0.0077 $cm^3/cm^3$) for the aerated concrete 600. The lowest values of this coefficient, 1.521% mass (0.0091 $cm^3/cm^3$), for the concrete 400 were recorded when measured with the non-invasive S2 sensor. In the case of the aerated concrete 600, the lowest value of this coefficient was achieved in the case of measurement with an invasive probe and was 1.579% mass (0.0063 $cm^3/cm^3$).

In turn, the maximum values of the RMSE coefficient for the cellular concrete 400 were obtained with the non-invasive S1 sensor (1.58% mass, 0.0162 $cm^3/cm^3$) and for the cellular concrete 600, again with the same sensor (1.93% mass, 0.0077 $cm^3/cm^3$).

According to the data presented by [49], the application of Topp et al. [49] in relation to selected soil centers resulted in uncertainties expressed as RMSE in the range of 0.01–0.066 $cm^3/cm^3$. In the case of the model proposed by Roth et al. [50], the RMSE ranged from 0.008 to 0.037 $cm^3/cm^3$. The RMSE for the popular calibration formula proposed by Malicki et al. (1996) was set at 0.03 $cm^3/cm^3$. It should be borne in mind that in most of the cited literature sources, the proposed models were universal. For this reason, the quality of the fit to the measured data was lower. The obtained RMSE values were comparable and in many cases lower than the values determined by the team of Udawatta et al. for regression models developed individually for each material (0.008–0.034 $cm^3/cm^3$) [51].

When analyzing the characteristics of the influence of moisture on the measurement uncertainty (Figure 5), it can also be noticed that the measurement uncertainties for the sensors S1 and S2 were smaller compared to those obtained for classic invasive probes. It was also found that the uncertainties read at low and high material moisture levels were greater than those obtained at middle moisture levels, which is a typical observation for many measuring devices. It also resulted from the adopted regression model [11,27].

In turn, in the case of the measurement uncertainty, it can be referred to the expanded uncertainty U. As its value is mainly influenced by the measurement uncertainty of type A, related to the quality of fit of the adopted regression model, the correlation between the U and the RMSE values was clearly visible. Only the U values from the middle measuring range were analyzed. The lowest value of expanded uncertainty was observed for the S2 sensor (concrete 400 class) corresponding to 1.799% mass. In the case of the measurement in concrete 600, the same values were also obtained with the S2 sensor, i.e., 2.468%mass.

Various literature sources indicate the following values of measurement uncertainties of models developed for soil media and invasive probes: according to [52] and [53], this uncertainty was in the range of 0.022–0.023 $cm^3/cm^3$; according to [54], it was 0.0269 $cm^3/cm^3$, according to [55], it was 0.004–0.018 $cm^3/cm^3$, and according to [50], it was 0.011–0.013 $cm^3/cm^3$. Most of these values are higher or comparable to those obtained in the course of the experimental research performed in this work.

## 5. Conclusions

The use of TDR surface sensors (noninvasive S1 and S2 type) showed that the Time Domain Reflectometry technique can be successfully used for non-invasive research to determine the moisture content of rigid porous materials used in construction. The TDR sensors provide very good responses similar to the measurements made with traditional invasive sensors.

Since, in contrast to the invasive probes, the electrodes of the TDR sensors do not require to be immersed into the tested material, the measurements are significantly simplified, and potential damage to both the sensor and the tested object is avoided.

When switching to the use of TDR sensors, however, one should be aware that they require individual calibration, which may be technically difficult to perform, especially in the case of partitions of objects, the characteristics of which are not known before starting the test.

It was also noticed that the RMSE of the tested sensor was higher for the almost dry and almost saturated states of the measured material. The obtained test results allow concluding that more accurate measurement results are achieved with the use of invasive sensors. The extended measurement uncertainty U(w) for measurements with an invasive probe (middle range of material moisture *w*) was equal to 2.955 [-] for the cellular concrete 400, while, for the cellular concrete 600, it was 2.897 [-]. In the case of the non-invasive sensors S1 it was for the 400 c.c. sample, 3.114 [-], and for the 600 c.c. sample, 2.889 [-]. In the case of S2 and the cellular concrete 400, it was 1.956 [-], and for the cellular concrete 600, it was 2.893 [-]. Finally, it should be stated that the expanded uncertainty for the FP/mts probe was 94.1% of that for S1 and 151% of that for S2 (cellular concrete 400); 99.72% of that for S1 and 99.86% of that for S2 (cellular concrete 600).

This does not change the fact that the surface sensors, despite the slightly different results compared to those obtained with the invasive probe, have a number of advantages. The following can be mentioned among them: (1) a simple and fast implementation allowing for more measurements per unit of time; (2) the time needed to prepare the device and the test object itself is much shorter than in the case of measurements with invasive probes; (3) the surface TDR sensors allow measuring the moisture of the tested objects without damaging them. The lack of need to introduce elements deep into the tested object allows maintaining the continuity of the structure of the object. Therefore, it is possible to measure fragile objects, e.g., objects with historic value or objects whose structure must not be destroyed.

**Author Contributions:** Z.S.: Conceptualization, methodology, data curation. M.P.-Ł.: software, investigation, writing—original draft preparation. M.W.-Ć.: visualization, funding acquisition. J.Z.: formal analysis, writing—review and editing. All authors have read and agreed to the published version of the manuscript.

**Funding:** This research was funded by Lublin University of Technology grant's number's M/KIRP/FD-20/IM-5/142/2022 and FD-20/IS-6/025.

**Institutional Review Board Statement:** Not applicable.

**Informed Consent Statement:** Not applicable.

**Data Availability Statement:** Not applicable.

**Conflicts of Interest:** The authors declare no conflict of interest.

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
