# Peer review of "Comparison of Measurement Possibilities by Non-Invasive Reflectometric Sensors and Invasive Probes"

_applsci, doi:10.3390/app13010665_

Round 1

Reviewer 1 Report

This manuscript reports a study comparing measurement possibilities of non-invasive reflectometric sensors with invasive probes. Unfortunately, the structure of the manuscript was not professionally written and arranged. Thus, it reduces the scientific merits of the manuscript. Typo needs to be carefully checked throughout the manuscript. Improving the English language is also recommended. Abbreviation needs to be carefully checked and defined throughout the manuscript. Results of Table 1-6 are repetitive with Figures 4-9. Why?. Results (Tables and Figures) were not presented in the scientific approach, did not reflect the title, and lack of novelty. The graph should be combined for easy discussion and reviewing process.

Author Response

Thank you for your thorough review. The answers to your questions are in the attached form required by the publisher.

Yours sincerely

Authors

Reviewer 2 Report

This is an interesting paper, aiming at vindicating the use of a non-invasive method of measurement of moisture in porous materials as compared with an invasive method. This may be of great interest in practical materials science applications. I have only several minor comments/suggestions that the authors should consider.

 1 Lines 20-1: In the sentence “The  use of the invasive TDR TM/mts probe gives more accurate test results compared to surface probes.” it would be useful to state quantitatively how more accurate the results are. 

 2 Can you add a version of Eq. (3) in which the derivatives are evaluated?

3.      Lines 284-5: It could be useful to put plots for 400 c.c. and 600 c.c. given now in Figs. 4-9 in the same graph for their better comparison.

4.      Line 317: It seems that in both Tables 8 and 9 the highest RMSE is for the S1 sensor.

5.      Units of quantities in Tables 8 and 9 are missing. The meanings of parameters p and df are not given.

6.      Fonts in plots are of rather different sizes and some are rather small.

7.      Some abbreviations (such as ‘FD’ or ‘FP/mts’) are not explained when they first appear in the text.

8.      Expressions like ‘5 × 5 × 12 cm’ are not correct and should be replaced by ‘5 cm × 5 cm × 12 cm’ or ‘(5 × 5 × 12) cm3’.

9.      The unit ‘%mass’ has ‘mass’ in too small fonts in several places in the text.

10.  Some references in the text are in bold and some are not. This should be unified.

11.  Line 360: ‘Root Mean Square Error’ should be move to an earlier place in the text where the abbreviation ‘RMSE’ is given for the first time.

12.  Typos: line 305 – ‘tables’ should read ‘Table’.

Author Response

Dear Reviewer,
Thank you for your thorough review. The answers to your questions are in the attached form required by the publisher.
Please see the attachment.

Yours sincerely
Authors

Round 2

Reviewer 1 Report

I am happy with the response made by the authors and the improved version of the manuscript is acceptable.